# Using a Machine Learning Approach to Evaluate the NOx Emissions in a Spark-Ignition Optical Engine

Federico Ricci, Luca Petrucci * and Francesco Mariani

Engineering Department, University of Perugia, Via Goffredo Duranti, 93, 06125 Perugia, Italy
* Correspondence: luca.petrucci89@gmail.com

**Abstract:** Currently, machine learning (ML) technologies are widely employed in the automotive field for determining physical quantities thanks to their ability to ensure lower computational costs and faster operations than traditional methods. Within this context, the present work shows the outcomes of forecasting activities on the prediction of pollutant emissions from engines using an artificial neural network technique. Tests on an optical access engine were conducted under lean mixture conditions, which is the direction in which automotive research is developing to meet the ever-stricter regulations on pollutant emissions. A NARX architecture was utilized to estimate the engine's nitrogen oxide emissions starting from in-cylinder pressure data and images of the flame front evolution recorded by a high-speed camera and elaborated through a Mask R-CNN technique. Based on the obtained results, the methodology's applicability to real situations, such as metal engines, was assessed using a sensitivity analysis presented in the second part of the work, which helped identify and quantify the most important input parameters for the nitrogen oxide forecast.

**Keywords:** machine learning; NARX; Mask R-CNN; emissions; engine



## 1. Introduction

Increasingly stringent pollutant emission standards and the fuel economy requirements put high demands on research into the efficiency of internal combustion engines (ICEs) [1,2]. OEMs are currently developing innovative strategies for future high-efficiency engines able to address this challenge, such as engine boosting and downsizing [3], low-temperature combustions (LTCs) [4], water injection [5,6] and lean [7,8] and/or EGR-diluted mixtures [9]. However, during the engine calibration process, the optimization of the efficiency and emissions requires engine parameters to be adjusted through extensive activities [10]. Moreover, the fine control of important operation variables can sometimes be hard to reach due to the inherent limitations of the measuring instruments [11,12].

Currently, machine learning (ML) approaches are widely used to solve problems in the automotive field, thanks to their ability to identify the intrinsic relationship between the input parameters and the engine response [13,14], with lower computational costs and faster operations than traditional methods [15,16]. ML algorithms demonstrated excellent results in predicting engine parameters such as pressure [17], fuel consumption [18], exhaust gas temperature [19], power [20] and emissions [21].

Considering the latter, Yaopeng Li et al. [21] employed an artificial neural network (ANN) with a genetic algorithm (GA) to optimize a direct dual fuel stratification (DDFS) strategy, starting from a numerical model of a light-duty diesel engine based on the General Motors 1.9 L platform. The optimized parameters (i.e., in-cylinder pressure and temperature, EGR rate, injection timing of fuels) were validated across a wide operating range. The performance was compared to that of a GA-CFD (computational fluid dynamics) approach. The ANN–GA method allowed improved fuel efficiency and lower nitrogen oxide (NOx) emissions to be obtained with lower computational time (over 75% of computational time saving). Samrendra K. Singh et al. [22] combined a genetic algorithm with a machine

learning technique called support vector regression (SVR) on a database of computational fluid dynamics simulations to develop a next-generation exhaust after-treatment system for diesel engines. The novel mixer design's main goal was to speed up the overall evaporation of the diesel emission fluid (DEF). While it was discovered that the evaporation rate predicted by the SVR and CFD (baseline) was within 7.5%, the suggested technique (CFD + SVR + GA) demonstrated an overall gain of 13.1%. Ruomiao Yang et al. [23] compared the performance of a random forest (RF) model and artificial neural network (ANN) in predicting the fuel consumption and emissions of a one-dimensional (1D) computational fluid dynamics (CFD) spark-ignition (SI) engine. To assess the performance of the established machine models, the engine performance in 2000 steady-state conditions was collected using a validated model at various spark timings (from $-40$ to 0 CA aTDC), engine speeds (from 1000 to 4000 rpm) and loads (from low- to high-level by adjusting the intake pressure from 0.5 to 1 bar). Both approaches were evaluated as able to assist the engine combustion analysis; however, ANN performed best, perhaps because the responses linked to engine combustion were better characterized by several interrelated mathematical functions.

Within this context, this work evaluates the possibility of applying the artificial neural network (ANN) technique to predict the pollutant emissions, i.e., nitrogen oxides (NOx), of an internal combustion engine (ICE). Tests were carried out on a single-cylinder spark-ignition (SI) engine with optical access at 1000 rpm and conditions of lean mixture, towards which automotive research is moving [24]. Starting from the internal in-cylinder pressure signals and the images of the flame front evolution captured using a high-speed camera, the aim is to compare the performance of the tested ANN architecture in predicting the NOx emission trend with the experimental data recorded using a fast NOx-$\lambda$ probe. In contrast to other methods, an artificial neural network that has been fine-tuned using experimental data may allow for the real-time evaluation of vehicle performance during running tests while also guaranteeing a greater level of input data reliability. Due to their close links to the investigated pollutant, both aforementioned variables were used to predict NOx [25]. The nitrogen oxide formation mechanism is a well-known mechanism that depends on three main factors, namely in-cylinder temperature, oxygen availability and residence time [25]. Unfortunately, the in-cylinder temperature of the burned zone cannot be experimentally estimated; however, the combustion speed and phasing suggest that the faster or the more advanced the combustion, the higher the peak in-cylinder pressure and temperature, which would augment the NOx rate of production. Through optical analysis, it is possible to gather extensive information on temperature and pressure rises and, consequently, on the synthesis of NOx by observing the formation and evolution of the flame front during the first stage of kernel formation.

The analysis of the flame front evolution was obtained by post-processing the grey-level images coming from a high-speed camera, whereas the in-cylinder pressure signal, coming from a piezoelectric transducer placed inside the engine chamber, was acquired using a fast combustion analysis system. The post-processing analysis was performed by using a Mask R-CNN (region-based convolutional neural network) approach [26,27], i.e., a convolutional neural network based on Faster R-CNN capable of detecting targets and performing semantic segmentation at the same time [28,29].

In a prior work of the same research group [30], the Mask R-CNN algorithm proved to be capable of detecting the kernel formation in advance and identifying combustions as regular rather than as anomalies, as in the case of other conventional approaches [31].

The NOx prediction was performed by a NARX (nonlinear autoregressive with external input) approach [32,33], i.e., a recurrent dynamic neural network used to model nonlinear dynamic systems and applied in time series [15,34].

In a previous work of the same research group [15], the forecasting performance of NARX was compared to that of FFANN [35,36]. In that work, the networks predicted the flow rate of GDI pumps intended for automotive applications. The results showed that the FFANN networks were not able to offer good predictions when the input data were closely related to the time component, while NARX was able to predict the time course of the

flow with greater precision. By reducing the input parameters to the model, i.e., excluding the less influential ones from the analysis, the predictive capabilities of NARX are also increased, thus leading to a significant reduction in the data that can be processed. Based on these considerations, NARX has been chosen as the method for the prediction of time series, i.e., NOx, in the present work. The results of this work showed the proposed model's ability to reproduce the experimental trend of the analyzed pollutant emissions. In particular, the prediction showed percentual errors always lower than 2%, with a maximum peak at about 1.6. The outcomes made it possible to thoroughly examine how the input factors affected the NOx forecast. For this reason, in the second part of the work, a sensitivity analysis using the Shapley value [37–39] was performed in order to explain the results, identify the most important input factors for the NOx prediction and assess the viability of using this methodology in practical settings.

## 2. Experimental Setup and Methods

### 2.1. Optical Access Engine

The single-cylinder research engine used to carry out the experimental campaign (Table 1) is a 500-cc with four valves, a pent-roof combustion chamber and port fuel injection system (PFI) [30]. It features optical access composed of a 45-degree mirror and a Bowditch piston with a 60 mm quartz crown (Figure 1), which allow light transmission in the visible range. The pressure levels inside the intake port and combustion chamber are recorded by a piezoresistive transducer (Kistler 4075A5) and a piezoelectric transducer (Kistler 6061 B, accuracy 0.5%), respectively. Both the corresponding signals, together with λ and NOx measured by a fast lambda probe Horiba MEXA-720 at the exhaust pipe (accuracy of $\pm 2.5\%$), are acquired by a Kistler Kibox combustion analysis system (temporal resolution of 0.1 CAD), which allows performing indicating analysis of the combustion processes. For each operating point tested, a total of 103 consecutive combustion events were recorded.

**Table 1.** Main features of the optical access engine.

| Displaced Volume | 500 cc |
|---|---|
| Stroke | 88 mm |
| Bore | 85 mm |
| Connecting Rod | 139 mm |
| Compression Ratio | 8.8:1 |
| Number of Valves | 4 |
| Exhaust Valve Open | 13 CAD bBDC |
| Exhaust Valve Close | 25 CAD aTDC |
| Inlet Valve Open | 20 CAD bTDC |
| Intake Valve Close | 24 CAD aBDC |

The mixture ignition was ensured by a barrier discharge igniter provided by Federal Mogul Powertrain—a Tenneco group company (Figure 2). Such an igniter was chosen to ignite the mixture since it proved to be capable of guaranteeing stable combustion processes in lean conditions with respect to a traditional spark [30].

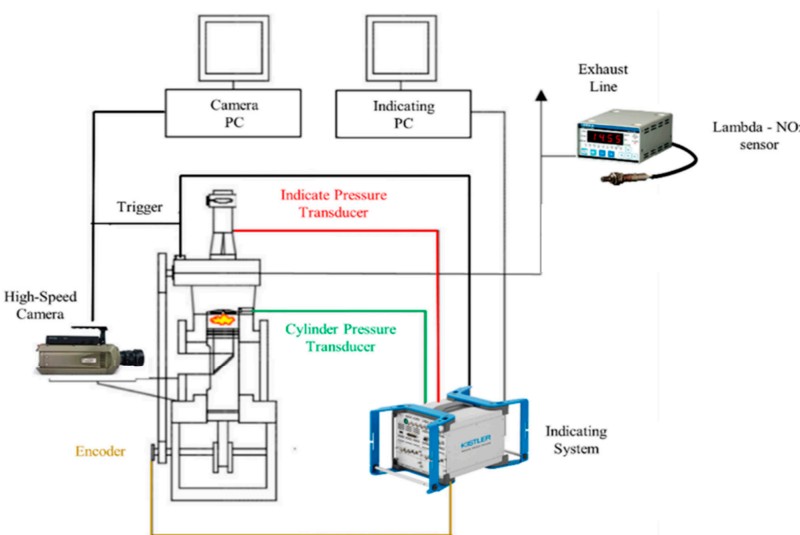

**Figure 1.** Experimental apparatus.

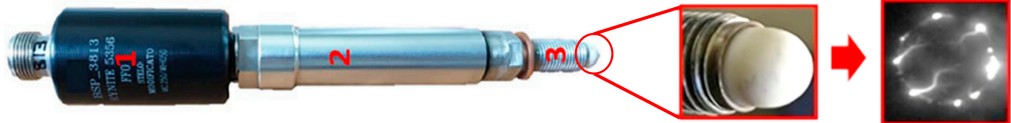

**Figure 2.** Tested igniter (**left**) (1 = inductor, 2 = connection, 3 = firing end) and corresponding firing end detail (**middle**). Discharge event (**right**): ionization waves (i.e., streamers) propagated above the igniter cupola.

### 2.2. Imaging System

The combustion process formation and evolution were captured using a Vision Research Phantom V710 high-speed CMOS camera coupled with a Nikon 55 mm f/2.8 [30]. The synchronization between imaging data and indicating ones, which is performed thanks to the signal (trigger) of an automotive camshaft position sensor (Bosch 0232103052), allowed matching the in-cylinder pressure trace of each process with the flame development 2D information (on a swirl plane) (Figure 3).

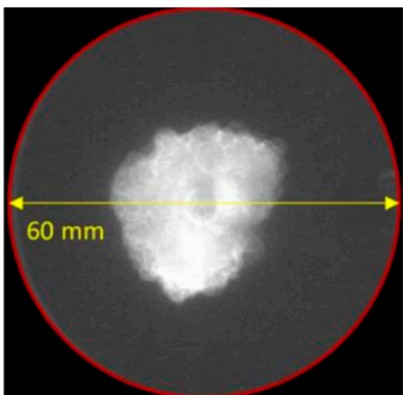

**Figure 3.** Detection of the flame front evolution.

The technical features of the high-speed camera are reported in Table 2. For each operating point tested, 63 consecutive combustions were recorded. Due to flame wrinkling, distortion and convection, the flame average radius, which can be detected without reaching the optical boundary, is about 20 mm, corresponding to about 5% of the mass fraction burned (MFB), detected by the indicating system.

**Table 2.** High-speed camera technical features.

| Feature | Value | Unit |
|---|---|---|
| Image resolution | $512 \times 512$ | pixel |
| Sampling rate | 20 | kHz |
| Exposure time | 49 | µs |
| Bit depth | 12 | Bit |
| Spatial resolution | 130 | µm/pixel |
| Temporal resolution (@1000 rpm) | 0.3 | CAD/frame |

*2.3. Test Campaign*

Tests were carried out on the optical access engine with the engine operating at 1000 rpm and low load (IMEP = 4.5 bar at λ = 1). The ignition timing was optimized for each operating point tested to achieve the maximum brake torque (MBT) (reached with the combustion center MFB50 around 9 CAD aTDC) [30]. For the sake of clarity, each tested point, listed in Table 3, was considered stable when presenting a $CoV_{IMEP}$ less than 4%. Three different kinds of λ were chosen since they present different in-cylinder pressure levels, and, therefore, diverse NOx emissions, and different levels of luminosity (decrease in brightness as the air/fuel ratio increases). This allowed for the evaluation of the tested architecture's predicting capabilities on a variety of NOx that are characteristic of combustions operating under lean conditions [31].

**Table 3.** Main features of the test campaign used as reference.

| λ, - | Ignition Timing, CAD aTDC | IMEP, Bar | $CoV_{IMEP}$, % |
|---|---|---|---|
| 1.3 | −22 | 3.42 | 1.32 |
| 1.4 | −26 | 3.19 | 1.04 |
| 1.5 | −38 | 2.95 | 1.95 |

## 3. Artificial Neural Network Setup and Methods

*3.1. Detection of the Flame Front Evolution*

The detection of the flame front evolution was performed by using a Mask R-CNN method [28,29], based on the images extracted from the experimental campaign of Table 3. Details on the neural network structure can be found in a previous work of the same research group [30]. In that work, the neural structure was trained and tested on images portraying the flame front evolution up to λ = 1.8. For this reason, the same weight found in the previous work was utilized for the present investigation. The 5th epoch of 10 was selected, and its weight was exported because it showed the best performance in terms of loss and values loss [40]. The binarization process required to determine the flame front evolution was directly realized on the obtained mask, as reported in Figure 4a–c. For each analyzed λ of Table 3, *n* images of *p* combustion events (Figure 5) were extracted from the high-speed camera and used to determine the $R_{eq}$. The images were selected starting from the end of the discharge (aEoD) up to the achievement of $R_{eq}$ equal to 30 mm. Without setting any binarization threshold, the binarization process was directly realized on the obtained mask, as mentioned before.

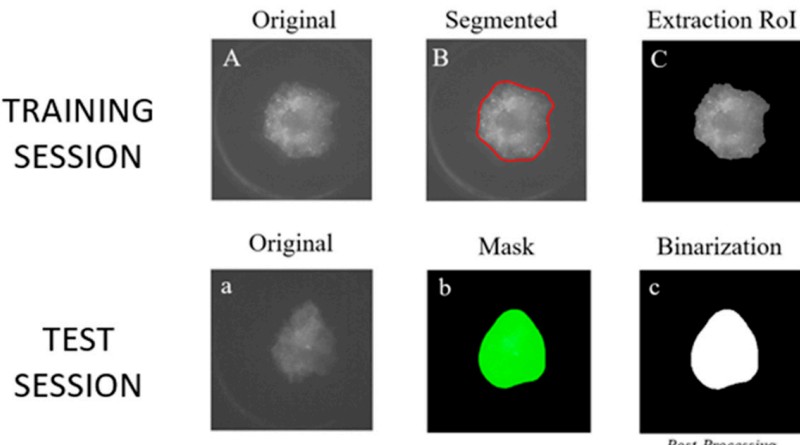

**Figure 4.** Extraction of the region of interest (RoI) during the training session ((**A**) = original image, (**B**) = segmentation of the raw image and (**C**) = extraction of RoI from the segmented image) and determination of the binarized image (**c**), starting from the mask process (**b**) applied on the original image (**a**). The binarized image is obtained during the test session, starting from the outputs of the training session [30].

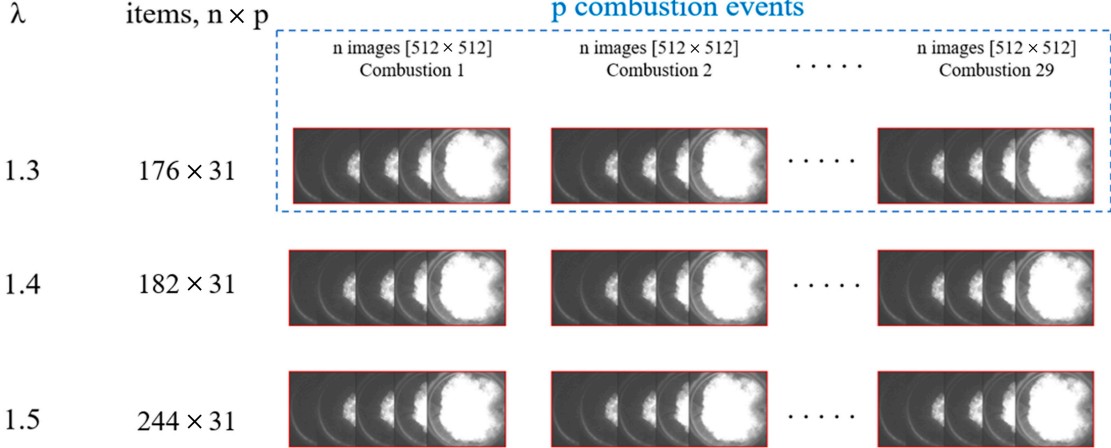

**Figure 5.** Number of items used for the training session at each lambda value analyzed.

The binarization process converts the grayscale images into black (unburned area with pixel values equal to 0) and white (burned area with pixel values equal to 1) ones to extract the equivalent flame area. The equivalent flame radius ($R_{eq}$) is defined as (Equation (1)):

$$R_{eq} = \sqrt{n_b * sc^2/\pi} \tag{1}$$

where $n_b$ is the number of white pixels and sc is the scaling factor (mm/pixel). Mask R-CNN automatically estimates the binarized area, without setting a defined threshold, thus allowing an analysis to be performed completely independently from the user interpretation.

### 3.2. Prediction of the NOx Emissions

The NOx prediction was performed using a NARX approach [32,33], i.e., a recurrent dynamic neural network used to model nonlinear dynamic systems and applied in time series modelling [15,34].

Such a network is composed of a series–parallel architecture (i.e., open-loop) or a parallel one (i.e., closed-loop) (Figure 6).

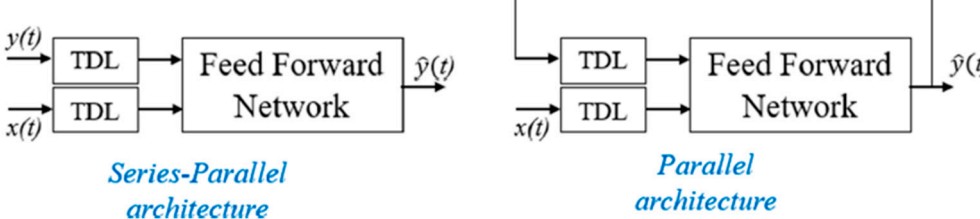

**Figure 6.** Architectures of the NARX neural network.

In the series–parallel architecture, the desired output value $\hat{y}(t)$ is predicted from the present and past values of the input *x(t)* and the true past value of the time series *y(t)*.

In the parallel architecture, the prediction is performed from the present and past values of *x(t)* and the predicted value of $\hat{y}(t)$.

A series–parallel architecture is used during the training phase because of the availability of the true past value of the time series. Then, the architecture is converted into a parallel one, useful for multi-step-ahead forecasting.

### 3.2.1. Time Series Analysis

First, Figure 7 reports the NOx emission trends, which have been continuously recorded (for a total of 103 consecutive events) by the λ-NOx probe on the optical engine, under the operating conditions shown in Table 3. NOxs show an increasing trend that is the more marked the richer the mixture.

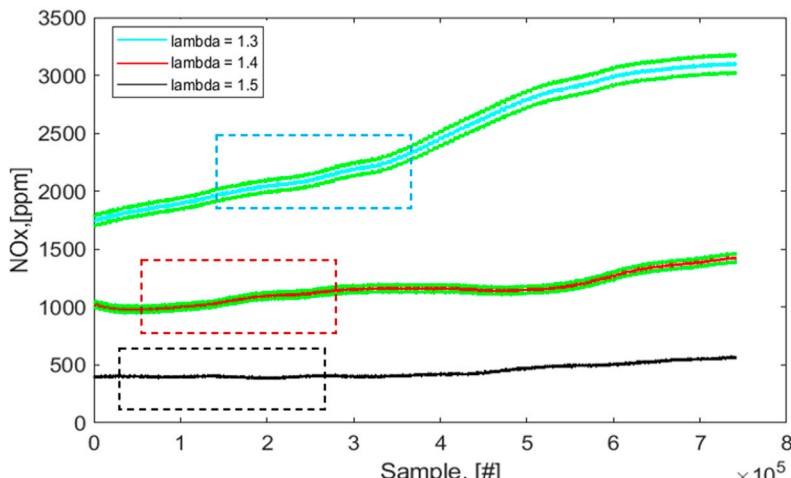

**Figure 7.** NOx trend of 103 consecutive experimentally acquired events on the optical access engine. The dashed boxes indicate the region of interest in which the NOx prediction was performed. The measurement error of each recorded point is represented by the green curves. The higher the recorded value, the higher the error committed. At λ = 1.5, the curve hinders the measurement error.

The NOx growth rate is due to the progressive increase of the in-cylinder temperature. Since such a quantity cannot be provided, Figure 8 displays the trends of the in-cylinder pressure, the distribution of the maximum in-cylinder pressure and the corresponding APmax (crank angle degrees at the maximum in-cylinder pressure). As can be observed, combustion occurs progressively earlier and the maximum peak reached in the chamber increases, thus leading to increments in the NOx. As mentioned, the richer the mixture, the higher the internal cylinder temperatures and, therefore, the NOx production and the corresponding growth rate. If there was the possibility to carry out tests of longer duration, as happens, for example, with metal engines (i.e., engines featuring cast iron piston rings instead of Teflon–graphite as in optical access engine), there would be a stabilization of the emission around an average value. In the optical engine, the intrinsic characteristics of the

system do not allow long-term tests to be performed. In any case, it is possible to observe that at the end of the acquisition, the emissions tend to stabilize around an average value. For this work, working areas (dashed boxes in Figure 7) far from the stabilization range were chosen in order to test the algorithm with data featuring high variability.

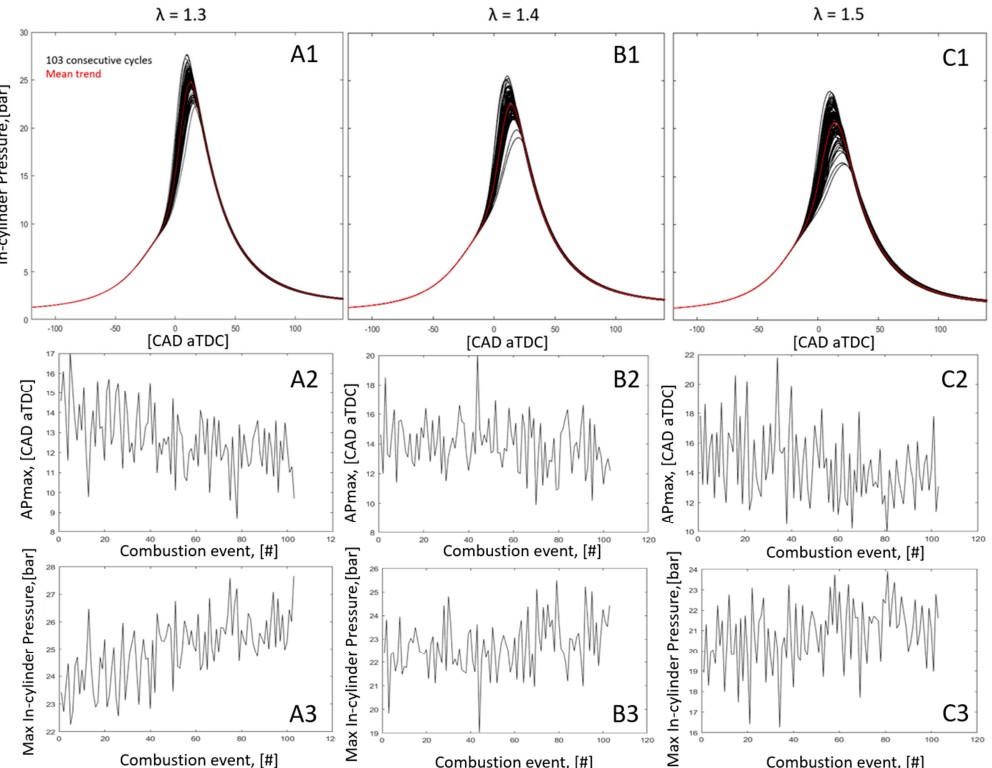

**Figure 8.** The trends of the in-cylinder pressure (**A1**–**C1**), the distribution of the maximum in-cylinder pressure (**A3**–**C3**) and the corresponding APmax (**A2**–**C2**). The curves have not been given with error bars because the in-cylinder pressure transducer's accuracy is quite low (see Section 2.1, Optical Access Engine).

In the first part of the present work, the neural architecture used to predict the NOx trend was composed of 2 hidden layers, each of which comprised 50 and 100 neurons, respectively (Figure 9). For each operating point analyzed (Table 3), 3 input parameters were chosen: equivalent flame radius (Equation (1)), in-cylinder pressure and NOx trend. Figure 10 reports an example of the trends of such quantities. For each λ, each parameter was characterized by the number of samples reported in Figure 4, corresponding to a sampling frequency of 20 kHz (or 0.3 CAD/sample at 1000 rpm). The training session was realized in the MATLAB environment on 28 consecutive combustion events, while the test session regarding the NOx prediction in 3 cases was different from the ones used for the training session. Figure 9 describes the neural network structure and the characteristics of the training and test session used to perform the NOx prediction.

The performance of the tested algorithms was evaluated in terms of RMSE. The root-mean-square error (RMSE) is frequently used to estimate the differences between the values predicted by a model (target) and the observed values (output); it is an estimator of the prediction quality [41]. The lower such a value, the better the estimation performed by the algorithm.

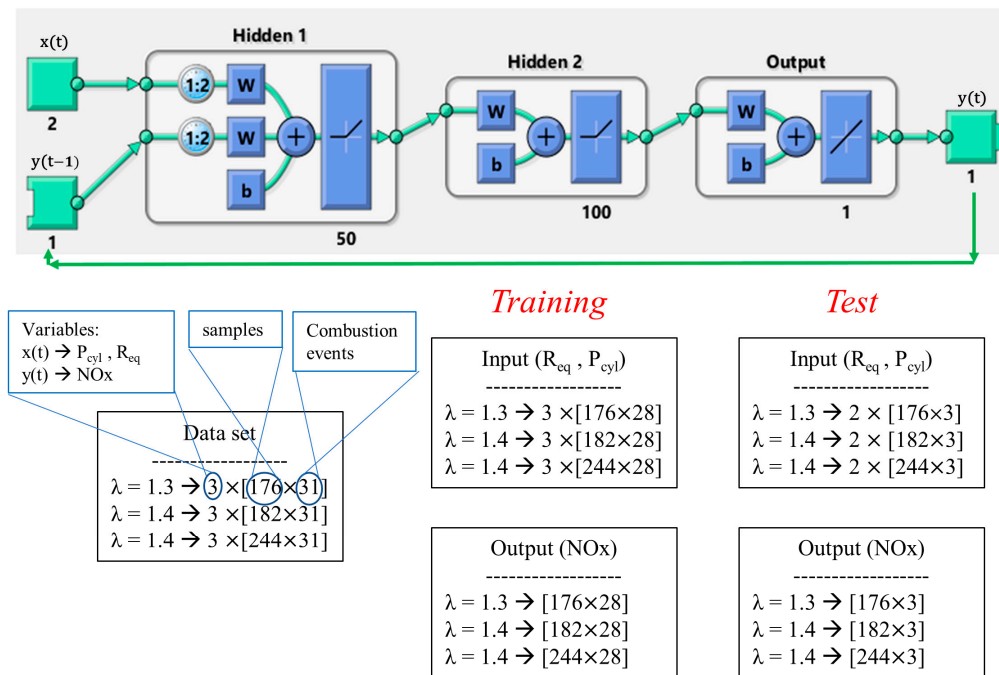

**Figure 9.** Neural network structure for the NOx prediction. Characteristics of training and test session.

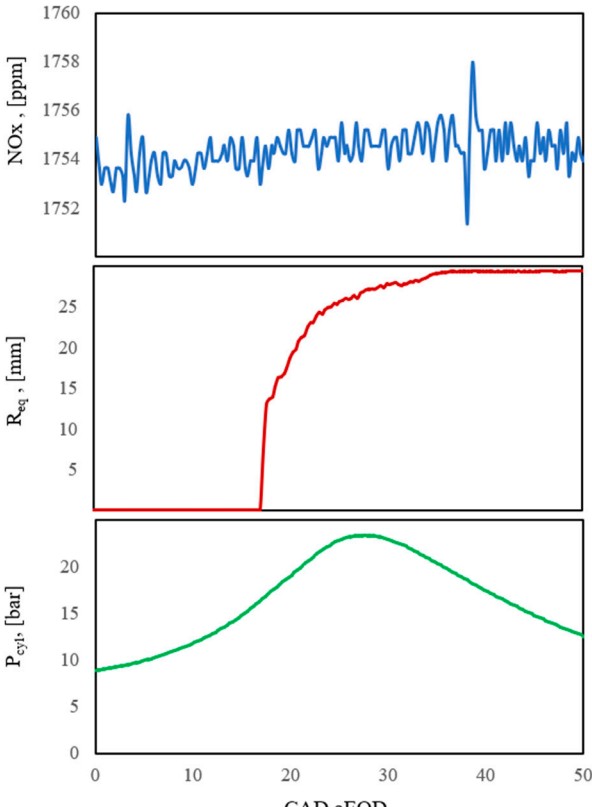

**Figure 10.** Examples of NOx emissions, equivalent flame radius and in-cylinder pressure experimental trends. The Pcyl curve has not been given with error bars because the sensor's accuracy is quite low (see Section 2.1, Optical Access Engine). The NOx curve has not been given with error bars because it would be difficult to discern the trend.

To sum up:

-   The Kibox analysis system provides indicating data of 103 consecutive combustion events, from $-360$ CAD aTDC to 359.9 CAD aTDC, with a temporal resolution of 0.1 CAD.
-   The high-speed camera provides the images of the flame front evolution of 63 out of 103 events with a sampling frequency of 0.3 CAD/frame at 1000 rpm.
-   According to the operator's decision, the high-speed camera starts recording once the trigger signal from the phase sensor is received.
-   To match the indication and image data of each event, the data from the Kibox were selected with the same sampling frequency as the high-speed camera, i.e., every 0.3 CAD starting from the end of discharge (EoD) to the achievement of the optical limit ($R_{eq}$ = 30 mm). This ensured that each image, i.e., $R_{eq}$, was matched to its own $P_{cyl}$ and NOx value.
-   A total of 31 of 63 events, in the range shown in Figure 7, were selected and the corresponding data (Figure 9) used as input for the neural architecture.
-   A total of 28 of these 31 events were used for the training session and the other 3 for the test.

### 3.2.2. Analysis of the Influence of the Input Parameters on the NOx Prediction

Based on the results obtained in the previous part of this work, a sensitivity analysis was carried out by using the SHAP library developed in the Phyton environment, specifically, the average absolute Shapley values (ABSV) [37,38]. The aim was to evaluate the impact of the single measured quantities (Figure 11) on the objective function (i.e., the average NOx emissions of the $i^{th}$ combustion event). The choice of the parameters of this analysis is strictly related to the input parameters of the previous analysis. Quantities connected to the in-cylinder pressure Pcyl, such as Pmax, APmax and IMEP and the equivalent flame radius $R_{eq}$, such as CAD aEoD$_{Req=9mm}$ and CAD aEoD$_{Req=20mm}$, were chosen. The latter variables indicate the crank angle degrees where the equivalent flame radius is equal to 9 and 20 mm, respectively. In particular, CAD aEoD$_{Req=20mm}$ can be associated with the 5% of mass fraction burned, as described in Section 2.1.

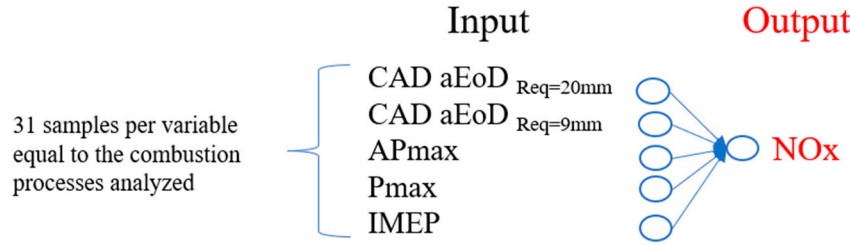

**Figure 11.** Description of the input parameters used to quantify their influence on the NOx prediction using the Shapley value.

### 4. Results and Discussions

Table 4 displays the quantities through which the forecasting performance of the NARX structure was evaluated. The standard deviation $\sigma$ of the observed series allows the evaluation of the variability of the target data, and it is used to understand the RMSE of the proposed neural architecture. Generally speaking, considering the specific $\lambda$ value, the lower the $\sigma$, the lower the RMSE. On average, for each $\lambda$, a higher RMSE value corresponds to observed data characterized by higher variability. In any case, the RMSE value is always lower than the acceptable threshold of 5 [15], thus highlighting the quality of the forecasting performance. In particular, it is worth highlighting that at $\lambda$ = 1.4, the RMSE of the analyzed series is close to the unit despite the highest $\sigma$ recorded. This occurrence could be related to the nature of the input parameters, namely Pcyl and the $R_{eq}$.

**Table 4.** Prediction quality evaluation, through RMSE, at each λ value.

| | $\sigma$ of the Observed Series | | | RMSE | | |
|---|---|---|---|---|---|---|
| | Series n.1 | Series n.2 | Series n.3 | Series n.1 | Series n.2 | Series n.3 |
| λ = 1.3 | 0.50 | 0.58 | 1.16 | 2.30 | 3.60 | 3.45 |
| λ = 1.4 | 0.62 | 0.89 | 0.83 | 0.97 | 0.96 | 1.02 |
| λ = 1.5 | 0.64 | 0.67 | 0.57 | 3.82 | 4.39 | 3.31 |

To correlate the quality of the NOx prediction with such quantities, Table 5 reports the mean values of the 28 consecutive combustion processes used for the training sessions, which are a function of Pcyl, namely $CoV_{\text{APmax}}$, $COV_{\text{Pmax}}$ and $CoV_{\text{IMEP}}$. Concerning the $CoV_{\text{IMEP}}$, since such a quantity is a function of the IMEP, expressed as $\oint Pcyl \times dV$, it could be helpful to also consider such a parameter to analyze the nature of the analyzed data. $CoV$ is expressed by the following relation (Equation (2)):

$$CoV = \frac{\sigma}{\mu} \tag{2}$$

that is, equal to the standard deviation ($\sigma$) over the mean value ($\mu$) of the analyzed quantities. In addition to the $CoV$ values, the standard deviations of the equivalent flame radius $\sigma_{\text{Req}}$ when $R_{\text{eq}}$ is equal to 9 and 20 mm are also considered. For sake of completeness, Figure 12 reports the equivalent flame radius of the combustion events analyzed at each λ value.

**Table 5.** Evaluation of the prediction quality by considering quantities related to the input parameters used for the training sessions.

| | $CoV_{\text{APmax}}$, [%] | $CoV_{\text{Pmax}}$, [%] | $CoV_{\text{IMEP}}$, [%] | $\sigma_{\text{Req=9mm}}$, [CAD aEoD] | $\sigma_{\text{Req=20mm}}$, [CAD aEoD] |
|---|---|---|---|---|---|
| λ = 1.3 | 10.96 | 8.15 | 1.19 | 1.34 | 1.32 |
| λ = 1.4 | 10.5 | 7.99 | 0.89 | 1.49 | 1.68 |
| λ = 1.5 | 18.91 | 11.8 | 1.32 | 3.43 | 2.97 |

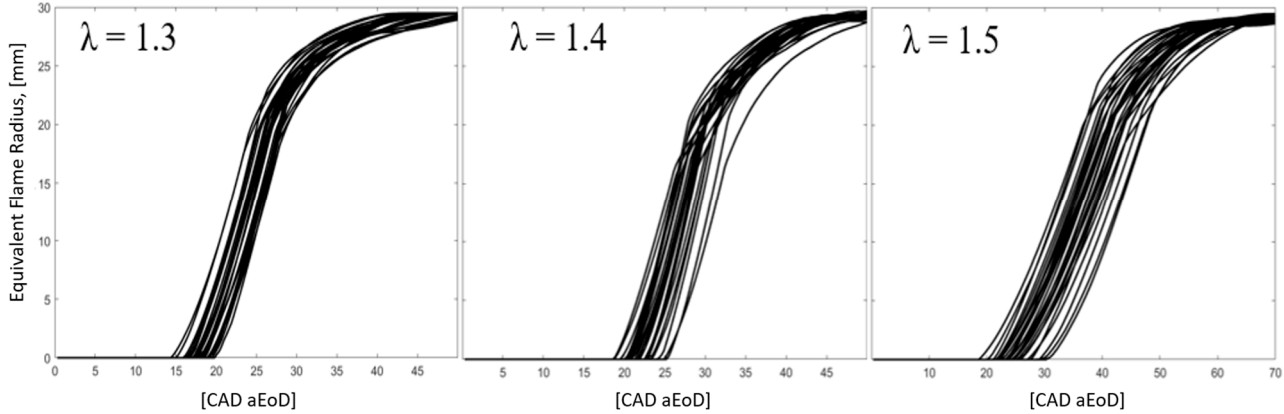

**Figure 12.** Equivalent flame radius of the cases used for the training session at each λ value analyzed.

From Table 5, it is possible to observe that the parameters connected to the in-cylinder pressure may influence the NOx prediction more than the ones related to the equivalent flame radius. In other words, the NOx prediction seems to be affected by the stability ($CoV_{\text{APmax}}$, $CoV_{\text{Pmax}}$, $CoV_{\text{IMEP}}$) of the process rather than the first part of the combustion formation and evolution ($\sigma_{\text{Req=9mm}}$, $\sigma_{\text{Req=20mm}}$). As a matter of fact, even if the resulting bundles are wider the leaner the mixture [30], the lowest RMSE recorded at λ = 1.4 could be related to the lowest value of $CoV_{\text{Pmax}}$, $CoV_{\text{IMEP}}$, while the highest RMSE at λ = 1.5 could be related to the highest values of such quantities. The lower influence of the first part of

the combustion can be expected, since the NOx production is more influenced by the part of combustion between 50 and 5% of the mass fraction burned.

Figure 13 shows the experimental predicted trends of all the cases analyzed, and Table 6 reports the deviation of the prediction from the target, namely the percentual error (%Err = (|Target − Predicted|/Target) × 100). From a qualitative point of view, the proposed model is able to reproduce the experimental trend. In particular, the prediction always shows a %Err lower than 2%, with maximum peak at %Err$_{max}$ = 1.61 (at λ = 1.5). The tested model can be considered as a valid alternative for the NOx prediction since the accuracy of the Horiba MEXA-720 used to record the NOx emissions is equal to 2.5%. In other words, the prediction is always lower than 65% of the measurement error of the fast NOx analyzer. The higher %Err shown at λ = 1.5 can be related to the higher variability of the observed data used for the training session.

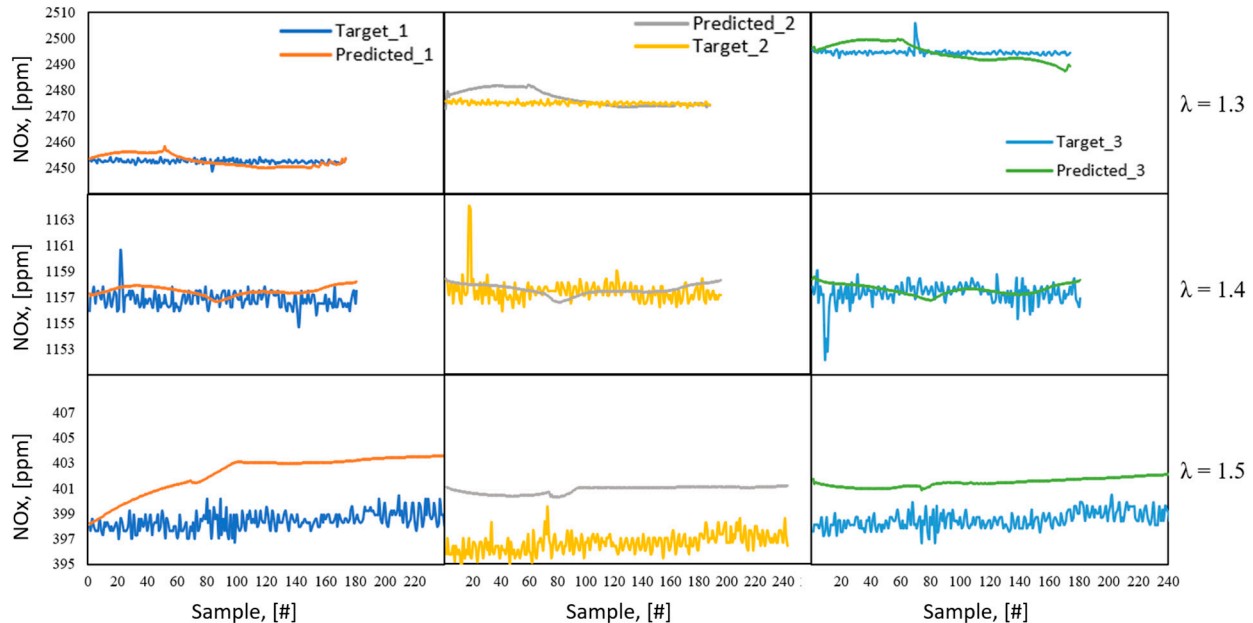

**Figure 13.** Experimental predicted trends of the predicted cases against the target ones at each λ value analyzed. The target curves have not been given with error bars because it would be difficult to emphasize the predicted trend.

**Table 6.** Percentual errors of the predicted cases at each λ value analyzed.

|  | %Err$_{max}$ | | | %Err$_{mean}$ | | |
|---|---|---|---|---|---|---|
|  | **Series n.1** | **Series n.2** | **Series n.3** | **Series n.1** | **Series n.2** | **Series n.3** |
| λ = 1.3 | 0.21 | 0.30 | 0.36 | 0.08 | 0.11 | 0.12 |
| λ = 1.4 | 0.24 | 0.52 | 0.52 | 0.07 | 0.06 | 0.06 |
| λ = 1.5 | 1.61 | 1.57 | 1.27 | 0.89 | 1.10 | 0.82 |

Based on the results of the previous analysis, Figure 14 shows the results of the sensitivity analysis carried out by means of the average absolute Shapley values. As is possible to observe, in all the λ cases analyzed, IMEP influences the NOx prediction more than the other input parameters. Furthermore, the parameters CAD aEoD$_{Req=9mm}$ and CAD aEoD$_{Req=20mm}$ are more influential than the ones related to the in-cylinder pressure, such as APmax and Pmax. Such a result confirms the prediction shown in the previous paragraph, thus testifying the greatest impact of IMEP on the NOx prediction and, at the same time, the right choice of input parameters connected to the equivalent flame radius. In any case, it is worth highlighting the impact of the other parameters related to P$_{cyl}$, i.e.,

APmax and Pmax. As a consequence of this, the methodology proposed in the previous paragraph could be exported to metal engines in which it is not possible to acquire images relating to the flame front evolution.

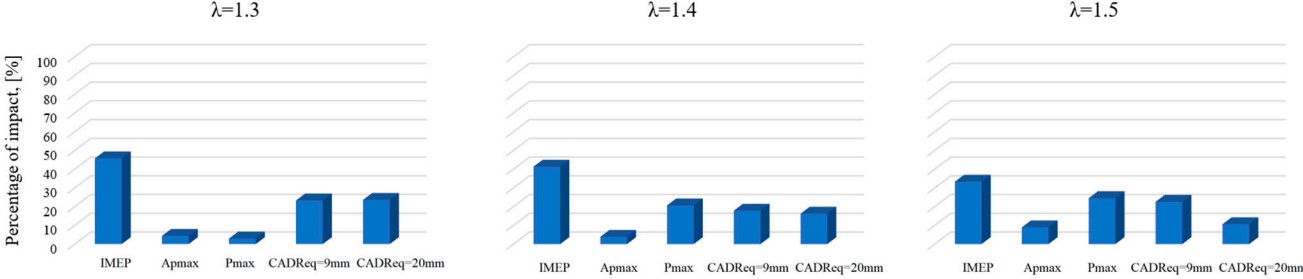

**Figure 14.** Global interpretation of the feature importance.

## 5. Conclusions

The present work investigates the possibility of implementing an artificial neural network technique to predict pollutant emissions, i.e., nitrogen oxides (NOx), of an internal combustion engine. Experimental activities were performed on a single-cylinder spark-ignition (SI) engine with optical access at 1000 rpm and conditions of lean mixture. The proposed approach was trained and tested on data coming from both indicating and imaging. The newly suggested method for assessing NOx emissions demonstrated the ability to replicate the experimental trend of the target variable, NOx concentration, while consistently maintaining a percentual error below 2%. The method showed consistent behavior in all engine operating situations, i.e., for the different values of the λ (excess air) index. The RMSE and *CoV* coefficient were used to assess the NARX structure. The RMSE value for each corresponds to observed data that are more variable, and in every instance, the RMSE value is less than the acceptable threshold of 5. The average absolute Shapley sensitivity study for all the operating conditions revealed that the IMEP quantity has the greatest impact on the forecasting model. The positive outcomes allow for the model to be tested and applied to a metal engine, allowing for a considerably wider variety of operating conditions to be investigated.

**Author Contributions:** Conceptualization, L.P. and F.R.; methodology, F.R.; software, L.P.; validation, L.P. and F.R.; formal analysis, L.P. and F.R.; investigation, F.R.; re-sources, F.M.; data curation, F.R.; writing—original draft preparation, F.R.; writing—review and editing, L.P.; visualization, L.P.; supervision, F.M.; project administration, F.M.; funding acquisition, F.M. All authors have read and agreed to the published version of the manuscript.

**Funding:** This research received no external funding.

**Data Availability Statement:** The data presented in this study are available from the corresponding author. The data are not publicly available due to privacy-related choices.

**Conflicts of Interest:** The authors declare no conflict of interest.

## Nomenclature

| | |
|---|---|
| %ERR | Percentage errors |
| ABSV | Absolute Shapley values |
| ACIS | Advanced corona ignition system. |
| aEoD | After end of discharge |
| ANN | Artificial neural network |
| APmax | Crank angle degrees at the maximum in-cylinder pressure |
| BDC | Bottom dead center |
| BDI | Barrier discharge igniter |
| CAD | Crank angle degree |

| | |
|---|---|
| CFD | Computational fluid dynamics |
| *CoV* | Coefficient of variation |
| CSI | Corona streamer (type of igniter) |
| ECU | Engine control unit |
| EGR | Exhaust gas recirculation |
| FFANN | Feed forward artificial neural network |
| GA | Genetic algorithm |
| GDI | Gasoline direct injection |
| ICE | Internal combustion engine |
| LTC | Low-temperature engine |
| IMEP | Indicated mean effective pressure |
| ML | Machine learning |
| IT | Ignition timing |
| MBT | Maximum brake torque |
| MFB | Mass fraction burned |
| MON | Motor octane number |
| NARX | Nonlinear autoregressive network with exogenous inputs |
| NOx | Nitrogen oxides |
| OEMs | Original equipment manufacturer |
| PAI | Plasma-assisted ignition |
| $P_{CYL}$ | In-cylinder pressure |
| PFI | Port Fuel Injection |
| R-CNN | Region-based convolutional neural network |
| $R_{eq}$ | Equivalent flame radius |
| RF | Radio frequency |
| RMSE | Root-mean-square error |
| RON | Research octane number |
| SI | Spark ignition |
| SVR | Support vector regression |
| TDC | Top dead center |
| ton | Activation time of the igniter |
| Vd | Driving voltage of the igniter |

Prefixes and suffixes

| | |
|---|---|
| a | after |
| b | before |
| cyl | cylinder |
| eq | equivalent |

Greek symbols

| | |
|---|---|
| λ | Excess air index |

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
