# Peer review of "Using a Machine Learning Approach to Evaluate the NOx Emissions in a Spark-Ignition Optical Engine"

_information, doi:10.3390/info14040224_

Round 1

Reviewer 1 Report

The paper presents the application of artificial neural network for the prediction of engine-out NOx emissions. The tests were performed on a single cylinder optical access engine, utilizing the measured in-cylinder pressure profiles and the images of flame front evolution to define the input parameters for the forecasting activities.

The results reproduced the experimentally obtained values of NOx emissions with an error under 2%, at three different air dilution levels. The research is extended by the sensitivity study to identify the influence of five input-related parameters on the NOx prediction. It was shown that the indicated mean effective pressure (IMEP) has the most influence on the forecasting model.

General comments:

Although the research topic is of high interest in the field of internal combustion engines and machine learning, the paper requires major improvements in both clarity and depth to increase it's readability and scientific value. Apart from the obvious minor drawbacks in terms of tables/figures formatting (font size, legend color not matching curve color etc.) and occasionally poor expressions, there are major concerns that need to be addressed.

The manuscript is not presented in a well-structured manner. The introduction section has a lot of lumped references without addressing the importance and relevance of each reference on the work performed in the paper. There are a couple of paragraphs not really important for the presented study, only unnecessarily expanding the introduction section and increasing the already high reference count. Instead, a more detailed discussion on a couple of most relevant references (application of neural network models for prediction of NOx) should've been made with the emphasis on methods used and the results achieved so far. For example, at least 5 relevant papers can be found in a matter of minutes only by searching the key-words „neural network“, „NOx prediction“ and „machine learning“, that deal exactly with this topic but none of them are mentioned in this manuscript. Analyzing the relevant work of others could perhaps explain the real novelty and identify a gap in knowledge that is to be fullfiled by the proposed work. The manuscript also includes an excessive number of unnecessary self-citations. For example, when the same method or equipment is mentioned at different sections in the manuscript, different reference of previous work by the authors is cited. A total of 15 or more self-citations can be easily reduced to 4 or 5 that are really relevant to this work. The experimental setup and methods section describes the equipment of the experimental test bed, measurements and postprocessing tools in too much detail since it is the same equipment used in prior work by the authors and already published. On the other hand, a more detailed description of the neural network structure for NOx prediction, training algorithm, the reasoning behind input parameters selection, as well as of training and test dataset selection is missing which is more important for this proposed study and would actually increase the clarity of the manuscript.

Most importantly, the intended usage of such model should be better explained for the presented method to be considered appropriate. Without explanation, the set of experimental results used for the evaluation of the proposed approach seems too scarce to support the presented conclusions. This is further elaborated in the detailed comments. Without clearly emphasizing the strengths and limitations of the model along with the intended use, the method itself is questionable. For example, there are publications where such models are used to predict the NOx emissions from the vehicle in real drive-cycles [i.e. Lee, J.; Kwon, S.; Kim, H.;Keel, J.; Yoon, T.; Lee, J. MachineLearning Applied to the NOxPrediction of Diesel Vehicle underReal Driving Cycle. Appl. Sci. 2021,11, 3758. https://doi.org/10.3390/app11093758 ]. This approach is clearly not intended for such use because of the selected inputs.

The detailed comments are given as follows:

·         Abstract:

1.       ... to identify the most influential input parameters for the nitrogen oxides prediction... This is misleading as the inputs to the model are restricted to those related to the in-cylinder pressure and images of flame front evolution.

2.       ... to evaluate the possibility of applying the methodology to real applications such as metal engines. It was not properly evaluated, only assumed by the authors that it could be feasible. The authors could've performed the same analysis with only in-cylinder pressure as model input and at least compare these two approaches.

·         Introduction:

3.       References [17-21] should be at least briefly discussed. For example, mention the model inputs, the range of validation dataset and the resulting accuracy. The same applies especially for reference [23].

4.       The sentence: Within this context, the herein work evaluates the possibility of applying the artificial neural network (ANN) technique to predict pollutant emissions.. should clearly state the only NOx emissions are in fact analyzed.

5.       The authors state: For the first time, to the best author’s knowledge, the forecasting activities were performed starting from data experimentally obtained. Is this considered as the main novelty of this work? Again, quick search pointed to two publications [Jongmyung Kim, Jihwan Park, Seunghyup Shin, Yongjoo Lee, Kyoungdoug Min, et al..Prediction of engine NOx for virtual sensor using deep neural network and genetic algorithm. Oil & Gas Science and Technology - Revue d’IFP Energies nouvelles, 2021, 76, pp.72. ff10.2516/ogst/2021054ff. ffhal-03445336f]. and [Mohammadhassani, J., et al. "Prediction of NOx emissions from a direct injection diesel engine using artificial neural network." Modelling and Simulation in Engineering 2012 (2012): 12-12.]. Nevertheless, it could be argued that the capability of such models to reproduce the reference results is not dependent on the source of the reference results.

6.       This entire paragraph is not very important for the proposed activity: Stable combustion processes at lean conditions can be realized, for instance, through the usage of innovative ignition systems [27-29]. For this reason, a Plasma Assisted Ignition (PAI) system [30], and in particular a Barrier Discharge Igniter [31,32], was chosen to ignite the mixture. Such an igniter proved to be capable of extending the lean stable limit of the engine [33-35] thanks to the generation of non-equilibrium (or non-thermal) plasma which accelerates the flame front evolution thanks to the combined action of thermal, kinetic, and transport effect [36,37]. The optical investigation allowed providing quantitative information about combustion inception and flame kernel evolution where the indicated analysis encounters its limit [38-41]. A lot of unnecessary references.

7.       The two paragraphs starting with: The analysis of the flame front evolution was obtained by post-processing... and ending with ... NARX has been chosen as the method for the prediction of time series: NOx in the present work.  would better fit to the METHODS section.

8.       The last paragraph in the introduction section represents the results and conclusions which should not be presented in the introduction.

·         EXPERIMENTAL SETUP AND METHODS

9.       Optical access engine, Igniter, and Imaging System are described in too much detail considering they are not used for the first time in this proposed work and are not crucial for the main scientific contribution of this work. They also mostly increase the number of unnecessary self-citations. They should be only briefly explained and referenced only with [54] as the entire setup is the same. The same Figures already used in the mentioned reference should also be avoided unless absolutely necessary. The focus should be on new methods or modifications to the previously used methods and models applied specifically for this work.

10.   Figure 1. has no real value, better to be replaced by the experimental setup scheme.

11.   Table 1., Table 2. and Table 3. -  the font is too small.

12.   The test campaign should include the explanation/justification on why such a small set is chosen to evaluate this method. For example, even if main set of only three dilution levels (engine loads) is kept, each of those could've been expanded with a couple of non-optimal spark timings. The combustion phasing affects the in-cylinder temperatures and as a result also significantly affect the NOx emissions.

·         ARTIFICIAL NEURAL NETWORK SETUP AND METHODS

13.   In the entire section, focus is once again on methods used to prepare the input parameters (i.e. flame front evolution detection and postprocessing). As this was already published, it should be only briefly explained and referenced. On the other hand, the entire description in the most important subsection (3.2 Prediction of the NOx emissions) is too brief. Although the architecture is said to be explained in detail in previous work [15], it is not clear if there are some modifications for this specific use. Assuming that the training and the resulting performance of such models should depend on the complexity of the problem, input parameters etc. this should be elaborated in detail, along with a brief theoretical background, characteristics, limitations etc. It could also be compared to the previous use presented in [15] performance-wise.

14.   Figure 6. – the legend color for lambda 1.4 does not match the curve color.

15.   In the subsection Time-series analysis the authors state: The NOx growth rate is due to the progressive increase of the in-cylinder temperature... and ... Having the possibility to carry out tests of longer duration as happens, for example, with metal engines, there would be a stabilization of the emission around an average value. In the optical engine, the intrinsic characteristics of the system do not allow to perform long-term tests. Why is this important? Does this mean that such model could not be used during transient operation, i.e. during the engine warm-up phase, and can be trained and used only on stable stationary operating points?

16.   Following up on the previous comment, the authors also state: For the herein work, working areas (dashed boxes in Figure 6) far from the stabilization range were chosen in order to test the algorithm with data featured by high variability. Is the assumption from the previous comment the reason why the entire measurement of 103 recorded engine cycles was not used?

17.   Additionally, the data sets are prepared for each excess air ratio as individual events. If a metal engine would be used, could a continuous measurement including the changes in operating conditions (i.e. engine load and engine speed changes) be fed to the model and what would be the outcome?

18.   Figure 7 – the font size is too small. Also, the unit on the x-axis for (combustion event, [°]) is wrong. IMEP values should also be reported in Figure 7, as the most influential input.

19.   The authors state: In the first part of the present work, the neural architecture used to predict the NOx trend is composed by 2 hidden layers, each of which comprised by 50 and 100 neurons, respectively (Figure 8). How is this determined and why? Is this exactly the same as in the previous application (ref. [15]).  

20.   It was really not clear from the start if the real input parameters used for the prediction are the crank resolved in-cylinder pressure and equivalent flame radius or the cycle-resolved values that are used later in the sensitivity analysis? From the presented results (Figure 12) it seems that each combustion event in the test phase is treated separately and the model indeed predicts the NOx emissions on a crank resolved basis. Why is this approach chosen? Between the selected 3 operating conditions, the values in the input vector prior to the combustion start would be almost identical. How does this affect the model? This again poses the question of how would the model predict the continuous change of NOx emissions due to operating conditions change if the input file would consist of large number of combustion events including the changes in operating conditions? Why not use the cycle-resolved values such as those used later in the sensitivity analysis in the first place? The prediction of crank-resolved NOx emissions in the exhaust does not have any additional value over the cycle-averaged value.

21.   In the subsection 3.2.2. Analysis of the influence of the input parameters on the NOx prediction, why are the values of 9mm and 20mm of equivalent flame radius chosen?

·         RESULTS AND DISCUSSIONS

22.   The font size in tables and figures is too small.

23.   Please explain this in more detail: In particular, it is worth highlighting that, at λ=1.4 the RMSE of the analyzed series is close to the unit despite the highest σ recorded. This occurrence could be related to the nature of the input parameters, namely Pcyl and the Req.

24.   Why is the influence of peak cylinder pressure position evaluated by the CoVApmax , and for the Req values by the standard deviation? Is CoVApmax really a good measure of cyclic variability of PCP position as its value is highly influenced by the actual average PCP position. Although not really possible in practice but imagine the actual PCP position being at TDC? Or, what if it is defined such that the position is not related to the FTDC but defined such that it is for example at 375°CA (FTDC being at 360°CA) Anyway, if the standard deviation of Pmax would be used instead how would this impact the results in Figure 13? What would happen if the standrad deviation for IMEP and Pmax is also used instead of CoV? The actual selection of input parameters for such analysis should be carefully defined and justified.

25.   The authors state: From Table 5 it is possible to observe that the parameters connected to the in-cylinder pressure may influence more the NOx prediction concerning the ones related to the equivalent flame radius. In other words, the NOx prediction seems to be affected by the stability (CovAPmax, CoVPmax, CoVIMEP) of the process rather than the first part of the combustion formation and evolution (σReq=9mm, σReq=20mm). The lower influence of the first part of the combustion can be an expected since the NOx production is more influenced by the part of combustion between 50 and 5 % of the Mass Fraction Burned [33]. This again poses the question of input selection. Why use an input that is known to not have a great influence on the objective quantity? Especially since the method and measuring equipment to obtain such input is very complex.

26.   The authors then state: Furthermore, the parameters CAD aEoDReq=9mm and CAD aEoDReq=20mm result to be more influent than the ones related to the in-cylinder pressure as APmax and Pmax. Such result confirm the prediction showed in the previous paragraph, thus testifying the greatest impact of IMEP on the NOx prediction and, at the same time, the right chose of input parameters connected to the equivalent flame radius. In any cases, it is worth highlighting the impact of the other parameters related to Pcyl, i.e. APmax and Pmax. As a consequence of this, the methodology proposed in the previous Paragraph could be exported to metal engines in which it is not possible to acquire images relating to the flame front evolution. This entire paragraph is contradictory, also opposite to the previous one. Please explain.

·         CONCLUSIONS

27.   The authors state: The method showed consistent behavior in all engine operating situations, i.e. for the different values of the λ (excess air) index. This conclusion cannot be backed-up by using only 3 operating points. Such scarce dataset could only indicate the possibility of consistent behaviour but should be further tested on a wide range of engine speeds, engine loads, spark sweeps etc.

28.   The encouraging results obtained provide the possibility of applying and testing the model to a metal engine with the possibility of investigating a much wider range of operating conditions. This is the assumption, not a conclusion based on the performed activities. The same procedure should've been applied with only in-cylinder pressure as the input and perhaps then, based on the obtained results, this could've been concluded.

Author Response

you can find attached the answers to your comments

Reviewer 2 Report

The manuscript uses machine learning to predict NOx emissions from a spark ignition engine. The manuscript is fairly well-written and it can be accepted after addressing the below mentioned issues:

1) There are many grammatical errors throughout the manuscript.

2) The authors have used abbreviations like NARX, CNN, APmax which has not be described.

3) Although the authors have used 68 references in the introduction section, but the section describes more on the work that the authors are doing instead of the current research in the field.

4) The novelty of the work is not clear.

5) The authors have used the word 'metal engine'. What does it refer to? Please add in the manuscript for the reader to understand

6) The authors have used equivalent radius from the optical images as input for the neural network. But there is no clarity on the relation between the radius and NOx emission.

7) In Fig. 5 the authors have mentioned 176 images for Lambda =1.3 but in Fig. 8 it is increased to 178. How?

8) In Fig. 8 in the training set NOx has been used as input and output as well. But not in test set. How is it possible?

9) Did the authors provide input for a combustion event for each crank angle degree as shown in Fig 9? Or did the authors take average value for each combustion event. If the values were taken for each crank angle then how did the authors measure NOx emission with respect to crank angle? Since no matter how fast the emission analyser is, it is extremely difficult to measure

10) The sentence "generally speaking, the lower the value of standard deviation the lower the RMSE" means something else but the Table 4 shows different results which is quite contradictory

11) The authors in Fig. 6 state that the NOx is considered for 103 consecutive events but the images are considered for 31 combustion events. The authors must clarify the procedure of acquiring the Req image and NOx data for the same combustion event.

Author Response

(The authors gave the same response as above.)

Reviewer 3 Report

In general, I consider that this subject is interesting, nevertheless, several issues need additional information, and some justifications must be added to improve the quality of the manuscript. It does not imply an outstanding innovation in the approach nor in the methodology in this research field, and the experimental procedure and the ML methodology are poorly explained. The contributions that this research makes compared to previously existing ones should be highlighted, and nothing new is found in your results. In this sense you should try to highlight what is the innovation of you approach, and focus the analysis on novelties, I am not sure if it is suitable for this journal, in more detail:

The explanations about the ML methodology are too poor, and the experimental procedure is not clearly explained too.

As in any experimental study, the measurement results should be shown with their analysis of uncertainty study included, in addition you should qualify the oscillations, standard deviation, fluctuations, etc. You must include the uncertainty study, please.

Some acronyms are not defined before it appears nor the nomenclature: OEMs, please review all the acronyms carefully.

Figure 6, legend colours do not agree with the plotted lines. Please modify properly.

Caption of Figure 6, ‘103’ number is repeated, please correct the mistake.

Caption of Figure 7 are not clear at all. There is not well defined de position and the reference of each subfigure, left, top, right. I recommend you to use some letters and more specific cross references.

There are some blue values marked in table 5, but no explanation has been included. Please clarify.

Figure 12. Please review the data carefully, in my opinion there is some change between predicted and target values, and/or figure legends.

One suggestion, in order to simplify the review process, I suggest you to activate the numbering of the lines of the document, since this way the reviewer could identify the location of the comment easier.

Author Response

(The authors gave the same response as above.)

Round 2

Reviewer 1 Report

Thank you for addressing each comment and clarifying some of the imposed concerns. The manuscript is sufficiently improved in terms of structure and presentation to be acceptable for publication. There are still some minor oversights remaining that I believe can be fixed in the final manuscript in the following publishing procedure and do not require another round of review.

1.       The references are not ordered correctly (i.e. the reference [28] is followed by the references [36,37]), then the references [45,16] are followed by [52-54]. Additionally, if not mistaken, the references [30], [32], [33], [48], [51] are not mentioned anywhere in the manuscript but remain on the reference list.

I will however use this opportunity to further elaborate on some of the comments that I believe were misunderstood or incorectly addressed, as well-intentioned suggestions for further research activities.

2.       The introduction section still has a lot of lumped references without addressing the importance and relevance of each reference on the work performed in the paper. This is not a major problem, but they cannot replace the references that should've been studied and cited. To repeat: For example, at least 5 relevant papers can be found in a matter of minutes only by searching the key-words „neural network“, „NOx prediction“ and „machine learning“, that deal exactly with this topic but none of them are mentioned in this manuscript. Analyzing the relevant work of others could perhaps explain the real novelty and identify a gap in knowledge that is to be fullfiled by the proposed work.  

3.       The manuscript still includes an excessive number of unnecessary self-citations. If not mistaken, a total of 18 self-citations is now only reduced to 16. In one of the previous works [Petrucci, Luca, et al. "Detecting the Flame Front Evolution in Spark-Ignition Engine under Lean Condition Using the Mask R-CNN Approach." Vehicles 4.4 (2022): 978-995.], the authors used the same experimental engine, the same igniter, the same equipment and method for imaging and detection of the flame front evolution. This can be used as one relevant self-citation when referencing each of theese aspects in the current manuscript. The other self-citation could be the work where the NARX approach was used for prediction of the flow rate of GDI pumps, to point the reader to where the details on the neural network structure can be found. That would probably be enough, 2 self-citations. A couple more might be tolerated, but 16 self-citations indicate an intention of self-citing all similar prior work without the actual necessity.

·         INTRODUCTION:

4.       Previous comment: The authors state: For the first time, to the best author’s knowledge, the forecasting activities were performed starting from data experimentally obtained. Is this considered as the main novelty of this work? Again, quick search pointed to two publications [Jongmyung Kim, Jihwan Park, Seunghyup Shin, Yongjoo Lee, Kyoungdoug Min, et al..Prediction of engine NOx for virtual sensor using deep neural network and genetic algorithm. Oil & Gas Science and Technology - Revue d’IFP Energies nouvelles, 2021, 76, pp.72. ff10.2516/ogst/2021054ff. ffhal-03445336f]. and [Mohammadhassani, J., et al. "Prediction of NOx emissions from a direct injection diesel engine using artificial neural network." Modelling and Simulation in Engineering 2012 (2012): 12-12.]. Nevertheless, it could be argued that the capability of such models to reproduce the reference results is not dependent on the source of the reference results.
The response: ...The experimental nature of the data is viewed as novel by the authors since an artificial architecture tuned on experimental data could allow for real-time evaluation of automobile amount during testing while also guaranteeing higher reliability of the input data in comparison to current techniques...
Elaboration: The authors fail to address the two mentioned publications that are also based on the real measured data, either from the experimental test-bed or the actual vehicle driving on the road, and can be used for the real-time prediction of NOx. Additionally, in the general comments I pointed to another publication where such models are used to predict the NOx emissions from the vehicle in real drive-cycles [i.e. Lee, J.; Kwon, S.; Kim, H.;Keel, J.; Yoon, T.; Lee, J. MachineLearning Applied to the NOxPrediction of Diesel Vehicle underReal Driving Cycle. Appl. Sci. 2021,11, 3758. https://doi.org/10.3390/app11093758].
As already said, the novelty and the contribution cannot be proven when the most relevant work of others is not included and commented in the literature review.

·         ARTIFICIAL NEURAL NETWORK SETUP AND METHODS

5.       Previous comment: Additionally, the data sets are prepared for each excess air ratio as individual events. If a metal engine would be used, could a continuous measurement including the changes in operating conditions (i.e. engine load and engine speed changes) be fed to the model and what would be the outcome?
The response: As the algorithm might be trained and evaluated using more input variables and a larger working range, for example, we expect positive results.
Further suggestion: I suggest that as a first step of further activity, you merge the 3 operating points (
λ=1.3, λ=1.4, λ=1.5) into a single input file of continuous combustion events and perform the test? This could serve as a fictional transient operation and either prove your assumption or at least indicate some problems.

6.       Previous comment: It was really not clear from the start if the real input parameters used for the prediction are the crank resolved in-cylinder pressure and equivalent flame radius or the cycle-resolved values that are used later in the sensitivity analysis? From the presented results (Figure 12) it seems that each combustion event in the test phase is treated separately and the model indeed predicts the NOx emissions on a crank resolved basis. Why is this approach chosen? Between the selected 3 operating conditions, the values in the input vector prior to the combustion start would be almost identical. How does this affect the model? This again poses the question of how would the model predict the continuous change of NOx emissions due to operating conditions change if the input file would consist of large number of combustion events including the changes in operating conditions? Why not use the cycle-resolved values such as those used later in the sensitivity analysis in the first place? The prediction of crank-resolved NOx emissions in the exhaust does not have any additional value over the cycle-averaged value.
Further elaboration:
The authors failed to address the most important question. Your approach is clearly different to those used in suggested references (see comment 3) because it uses crank-resolved values as input rather than the cycle-resolved values. This is the novelty that the authors should've focused on to justify why such approach is better, if it is better. Since the intention is still to predict the NOx emissions in exhaust pipe (which are not crank-angle dependant), and not, for example, the formation of NOx in the cylinder (which is crank-angle dependant), it is hard to see any benefit. The Figure 10 in the revised manuscript is the perfect example (the inputs are crank-angle dependant and the output is not). How is this not problematic for the model?

·         RESULTS AND DISCUSSIONS

7.       Previous comment: Why is the influence of peak cylinder pressure position evaluated by the CoVApmax , and for the Req values by the standard deviation? Is CoVApmax really a good measure of cyclic variability of PCP position as its value is highly influenced by the actual average PCP position. Although not really possible in practice but imagine the actual PCP position being at TDC? Or, what if it is defined such that the position is not related to the FTDC but defined such that it is for example at 375°CA (FTDC being at 360°CA). Anyway, if the standard deviation of APmax would be used instead how would this impact the results in Figure 13? What would happen if the standrad deviation for IMEP and Pmax is also used instead of CoV? The actual selection of input parameters for such analysis should be carefully defined and justified.
The response: Since the CoVs are frequently employed in the automotive research sector to measure the cycle-to-cycle variance, the authors used these parameters as input rather than the standard deviation. Since training and test sessions would be conducted using parameters linked to the CoVs in any situation, the authors conclude that the choice of the standard deviation of these quantities could not have an impact on the prediction of the architecture. Furthermore, CoV and standard deviation are strictly connected since CoV is proportional to σ.
Further elaboration: I mistakenly assumed that the inputs to the sensitivity analysis where in fact the CoVs and not the actual values of IMEP, Pmax and Apmax.
The question however remains, why calculate and show the CoV of peak pressure position, but standard deviation of inputs related to the flame front development. The CoVs are indeed employed in automotive research sector to measure cyclic variability, but not very often do I see the CoV of crank angle related indicators such as peak pressure position or CA50. It cannot give very reasonable insight on cyclic variability because the CoV value would be highly dependant on the actual average position. For example, advancing the combustion phasing can improve the combustion stability, decreasing the CoV(IMEP) as well as CoV(Pmax) but by shifting the average position of Pmax closer to TDC the CoV(Apmax) would start to increase. Consider the CoV of the extreme fictional case I presented (average Apmax coming towards 0°CA ATDC). CoV would become infinite.

·         CONCLUSIONS

8.       It still seems that by extending the dataset used for training and tests, or the model inputs, entirely different conclusions could be obtained. For example, the sensitivity study identified IMEP as the most influential parameter that affects NOx prediction. However, the new dataset could be created by enforcing the fixed IMEP value (by intake throttling if not possible otherwise) at different excess air ratios, spark timings etc., all significantly affecting the NOx emissions. Additionally, the same value of NOx concentration in the exhaust can be obtained at very different values of IMEP. How is this possible if IMEP is the most influential parameter? The sensitivity study would make more sense in future activities as the authors stated: In real engine applications, you can use more parameters to predict NOx emissions.  Future research will undoubtedly concentrate on assessing the performance of neural networks utilizing large and reduced datasets or just the in-cylinder pressure as an input.

Author Response

The authors thank the reviewer for their judgments and recommendations. Attached is the response to the comments.

Reviewer 2 Report

The manuscript has been profusely revised and it can now be accepted

Author Response

The authors thank the reviewer for their judgments and recommendations.

Reviewer 3 Report

First of all, I would like to acknowledge the effort made by the authors in the detailed and extensive answers, and the effort to include some modifications and clarifications in the document. I consider that the quality of the paper has been improved, and now it could be published.

Author Response

(The authors gave the same response as above.)
